


# A synthetic satellite dataset of *E. huxleyi* spatio-temporal distributions and their impacts on Arctic and Subarctic marine environments (1998-2016)

Dmitry Kondrik[1], Eduard Kazakov[1], Dmitry Pozdnyakov[1]

[1]Nansen International Environmental and Remote Sensing Centre, Saint Petersburg, 199034, Russian Federation

*Correspondence to*: Eduard Kazakov (ekazakov@niersc.spb.ru)

**Abstract.** A 19-year (1998-2016) continuous dataset of coccolithophore *E. huxleyi* distributions and activity in Arctic and Subarctic seas is presented. The dataset is based on optical remote sensing data (mostly OC CCI data) with assimilation of different relevant in-situ observations, preprocessed with authorial algorithms. Alongside with bloom locations, we also provide both detailed information on *E. huxleyi* impacts within the bloom area on marine environments and the subdatasets of quantified coccolith concentrations, particulate inorganic carbon content and $CO_2$ partial pressure in water driven by coccolithophores. All data are presented on a regular 4x4 km grid at a temporal resolution of 8 days. The paper describes the theoretical and methodological basis for all processing and modeling steps. The data are available on Zenodo: https://doi.org/10.5281/zenodo.1402033.

## 1 Introduction

Ongoing climate change is a background of numerous emerging hot topics. Among them, alterations of both biodiversity in marine environments and the carbon balance in the atmosphere-ocean system (Rost et al., 2008). In some specific cases both processes are interrelated being spurred up by one and the same agent(s). Along with other marine inhabitants, coccolithophores are such entities, and more specifically, the algal species named *Emiliania huxleyi* – a unicellular planktonic organism that is most widespread in the world's oceans. Being simultaneously a calcifying and photosynthetic primary producer of, respectively, inorganic and organic carbon, *E. huxleyi*, in the course of its life cycle, enhances both the concentration of calcite and carbon dioxide partial pressure in ocean surface water. At least within *E. huxleyi* bloom areas, both processes are capable of changing the carbon balance, and hence affect both $CO_2$ fluxes between the atmosphere and surface ocean and the aquatic biogeochemistry. Being a spatially huge phenomenon invariably occurring in both hemispheres, and gradually propagating in the poleward direction due to $CO_2$ accumulation in the atmosphere (Rivero-Calle et al., 2015) and ensuing climate warming (Johannessen, 2008), *E. huxleyi* blooms are believed to be highly relevant to understanding the comprehensive nature of the changes unfolding on our planet.

Historically, the initial building up of knowledge on coccolithophores in general and *E. huxleyi*, specifically, was broadly based on in situ approaches effected in the course of both shipborne and laboratory activities. Extensive data were obtained



on *E. huxleyi* cell morphometry, internal structure, intracellular dark – and photoreactions, factors controlling/affecting the cell growth, as well as intrinsic optical properties, such as sun light total and spectral absorption, scattering/backscattering (Balch et al., 1996a). In addition, regression relationships were established between *E. huxleyi*-driven changes in both inherent hydro-optical parameters and $CO_2$ partial pressure in surface water within the bloom area (Holligan et al. 1993).

However, as this phenomenon extends over marine areas in excess of hundreds of thousand square kilometres (Balch et al., 2016; Kondrik et al., 2018a), and is spatially and temporally highly dynamic, solely satellite remote sensing approach means are able to comply with the challenge of studying it.

Until recently, only few satellite studies were performed and published on the typical locations of *E. huxleyi* blooms and associated concentrations of particulate inorganic carbon in surface ocean within the bloom area (e.g. Gordon et al., 2001;

Balch et al., 2016).

Prior to the publication by Kondrik et al. (2018a), no attempts have been undertaken to either retrieve from spaceborne data the total content of inorganic carbon produced by a *E. huxleyi* bloom (PIC) and increase in $CO_2$ partial pressure ($\Delta p CO_2$) in surface water within the bloom area or else reveal intraannual and interannual variations in the location and intensity of *E. huxleyi* blooms. No concatenated time series data are available to date on the associated quantifications of bloom surface,

bloom intensity, $\Delta p CO_2$ for all *E. huxleyi* blooms occurring within extensive latitudinal belts and encompassing waters of different oceans i.e. marine tracts significantly distanced longitudinally.

Meanwhile, the above specified information is an indispensable step towards a further pan-global inventory of the effects produced by *E. huxleyi* blooms on both marine chemistry and ecology, and $CO_2$ exchange fluxes between the atmosphere and ocean as such fluxes condition the status of the world's oceans as a sink of $CO_2$.

Based on the employed spaceborne ocean colour information, the present paper reports on extensive concatenated original datasets generated for subpolar and polar seas of the Northern Hemisphere, viz. North, Labrador (with adjacent North Atlantic open waters), Norwegian, Barents, Greenland and Bering seas. The obtained datasets are processed into a nearly two decadal (1918-2016) time series for each of the target seas/marine areas.

The collected data base of PIC and $\Delta p CO_2$ values in surface water within the bloom area together with intraannual and

25 interannual variations in the location and intensity of *E. huxleyi* blooms over such a variety of seas and across a nearly 20-year time period is presently unique.

Conjoined with a wealth of presently available supplementary data from satellite and shipborne missions on the environmental conditions under which target *E. huxleyi* blooms emerged and developed, the synthetic dataset we are reporting herein opens the way to detailed analysis of forward and feedback mechanisms governing the temporal and spatial

dynamics of this phenomenon. Further utilization of the results of such analysis in regional and global climatic models promises to predict future directions of development of the phenomenon in question (Rost et al., 2008).



## 2 Methodology and dataset content

Based on the facility of available satellite OC CCI and SeaWiFS data in the visible part of the spectrum, the following products have been generated to achieve the goals specified in the previous section, viz.: 1. *E. huxleyi* bloom extent; 2. Concentration of coccoliths within the bloom; 3. Total content of particulate inorganic carbon (PIC) produced by the bloom;

4. Increase in $CO_2$ partial pressurein marine surface waters due to the blooming phenomenon.

### 2.1 Bloom area quantification

Quantification of *E. huxleyi* bloom areas was performed in two stages. Firstly, RGB (red-green-blue) images were generated based on the weighted remote sensing reflectance, $R_{rs}$, which is the upwelling spectral radiance just above the water–air interface normalized to the downwelling spectral irradiance at the same level (Bukata et al., 1995). $R_{rs}$ values in the channels

centered at 670, 555, and 443 nm were employed. Analysis of the spaceborne radiometric data collected by Kondrik et al. (2017a, b) from the 5 target seas, yielded statistically robust specific ranges of $R_{rs}(\lambda)$ highlighting *E. huxleyi* blooms as turquoise areas; the areas of blooms of other (noncalcifying) algae were reflected in the images as green. Areas with scarce noncalcifying algae abundance showed up as blue or dark blue. The land mask was overlaid so that land areas were coloured light yellow.

In the second stage of quantification of *E. huxleyi* bloom extent, an additional criterion was imposed on the revealed turquoise areas: $R_{rs}$ values should be maximal at 490 nm and/or 510 nm, while at other wavelengths they need to be in excess of 0.001 (412 nm), 0.008 (443 nm), 0.01 (490 nm), 0.008 (510 nm), 0.008 (555 nm), and ~0 (670 nm). Such a selection provided the highest accuracy of bloom delineation. With the known pixel size, the bloom area can be confidently quantified. An example of *E. huxleyi* bloom extent masking is shown in Figure1.

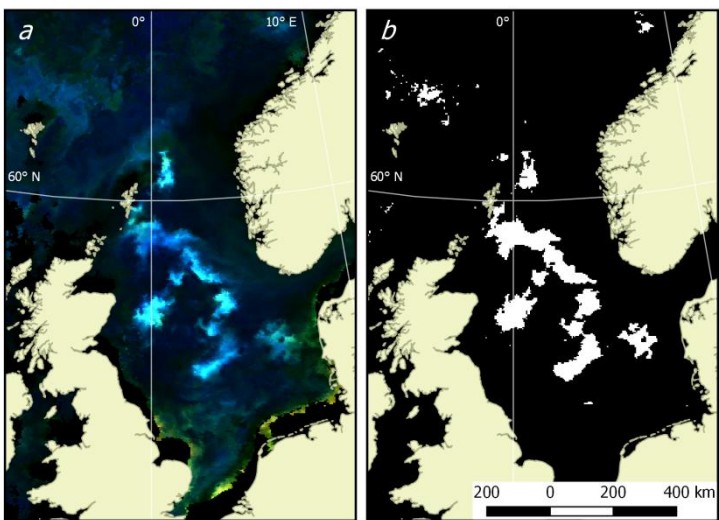



**Figure 1: Example of the bloom masking algorithm performance. *a*= source of the OC CCI RGB imagery for the North Sea (2016.06.09, with land mask); *b* = calculated bloom mask (white pixels stand for bloom detected, black pixels are areas void of bloom).**

## 2.2 Determination of the coccolith concentration

Determination of the coccolith concentration within the bloom was performed with the BOREALI algorithm (Bio-Optical REtrieval ALgorIthm, Korosov et al. 2009), based on the Levenberg–Marquardt (L-M) finite difference technique (Press et al. 1992). The L-M technique solves the inverse problem, i.e. in our case allows to retrievethe concentrations of water constituents from spectral subsurface remote-sensing reflectance, $R_{rsw}(\lambda)$, which is the upwelling spectral radiance just beneath the water–air interface normalized to the downwelling spectral irradiance at the same level (Jerome et al., 1996). A hydro-optical model accommodating spectral specific absorption and backscattering coefficients of *E. huxleyi* cells and coccoliths as well as pure water per se, non-calcifying alga and dissolved organic matter was developed and employed to run the BOREALI (Kondrik et al., 2017a).

The results of validation of coccolith concentration retrievals with BOREALI were assessed through the following statistical measures: coefficient of correlation, $r$, linear regression equation, $f(x)$, coefficient of determination, $R^2$, root mean square deviation/error, RMSE, systematic error, BIAS, and MAE. BIAS and MAE were then also normalized to the absolute values of coccoliths concentrations determined by using each model: $r = 0.88$; $f(x) = 0.6159x + 6.9197$; $R^2 = 0.77$; RMSE = $3.55 \times 10^9$ coccoliths·m$^{-3}$; BIAS = 25.30%; MAE = 32.30%.

In addition, ascertained by both RGB and $R_{rs}$ approaches, *E. huxleyi* bloom areas were further checked up using the results of coccolith concentration retrievals. This was done through the application of a threshold. A threshold of $90 \times 10^9$ coccoliths·m$^{-3}$ was chosen because, firstly, it assures the best correspondence between the bloom surfaces, determined by our radiometric and BOREALI algorithms. Secondly, this threshold is very close to the average value of coccolith concentrations in developed *E. huxleyi* blooms reported from the world's oceans (for references, see Balch et al. 1996b; Balch et al. 2005). The numerical assessments of bloom surfaces delineated/quantified by above independent ways converged precisely.

## 2.3 Coccolith content, particulate inorganic carbon and CO$_2$ partial pressure increment determination

Determination of the coccolith content (CC) was performed through establishing mixed layer depth (MLD) within the bloom area. The climatology of Montegut et al. (2004) was applied. The identified areas of *E. huxleyi* blooms with retrieved concentrations of coccoliths were overlapped by the respective climatological MLD fields, and for each pixel, the value of MLD was further used for calculating CC. Further, CC values were used to quantify the total content of particulate inorganic carbon (PIC). It was done for each 8-day time period (corresponding to the temporal resolution of the spaceborne radiometric data employed) through multiplying the carbon mass per coccolith, $m$, and CC followed by summarizing the results of multiplication within all pixels of respective bloom extent. The value of $m$ was equalled to 0.2 pg (Balch et al., 2005). The moment, at which the PIC assessment could be ideally performed in each bloom, corresponded to the situation when two conditions were fulfilled: (a) the bloom attained its largest surface, and (b) the spectral curvature of remote sensing



reflectance, $R_{rs}(\lambda)$, exhibited a maximum at about 490 nm as the location of $R_{rs}$ maximum at about 490 nm is an indication that the bloom is prevalently composed of coccoliths (Kondrik et al., 2017a).

Remote determinations of *E. huxleyi*-driven $pCO_2$ increment ($\Delta pCO_2$) consisted in establishing a relationship between *E. huxley*i-driven changes in $pCO_2$, that is, $\Delta pCO_2$, in bloom pixels, and the respective values of $R_{rs}$ (490). Such a relationship
5   (Kondrik et al., 2018a) with the following statistical characteristics: coefficient of determination, $r^2$ = 0.54, p ≪0.001, and RMSE = 23.4 µatm was used to quantify the spatial variations of $\Delta pCO_2$ in the target seas followed by recalculating $\Delta pCO_2$ for the water temperatures (retrieved from spaceborne data) that actually occurred during respective *E. huxleyi* bloom events (Copin-Montegut, 1988).

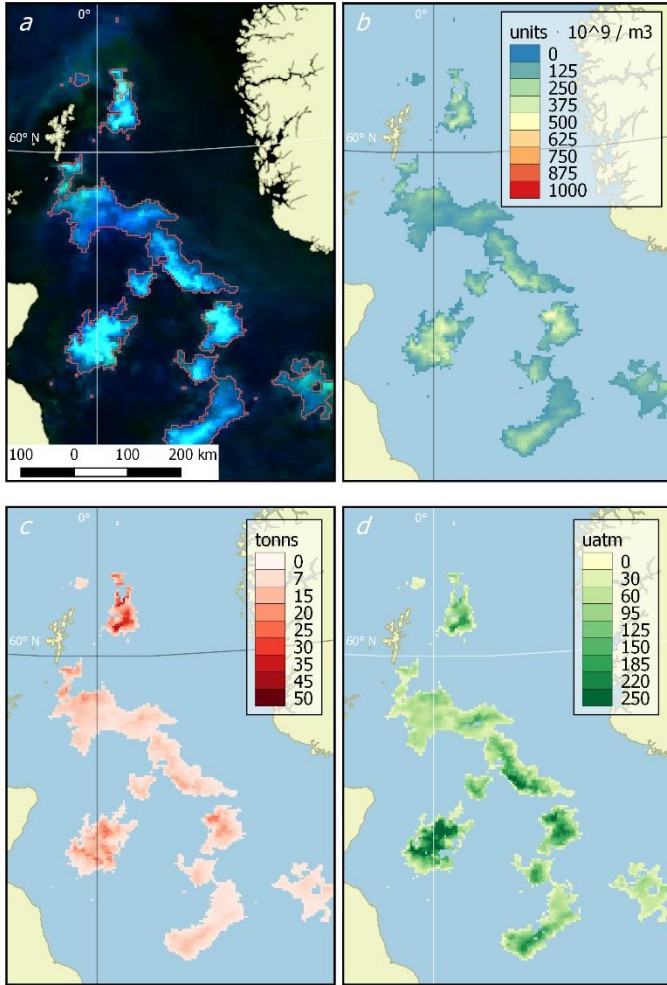

10  **Figure 2: Example of dataset products (the North Sea, 2016.06.09). *a* = source OC CCI RGB imagery with the bloom mask contoured in red, *b* = coccolith concentration ($10^9 \cdot m^{-3}$), *c* = content of particulate inorganic carbon (tonns), *d* = increase in CO$_2$ partial pressure in water (µatm).**



## 2.4 Additional technical workflow

In the cause of satellite data processing, several preceding procedures were performed.

1. Reprojection of satellite images. Given the high latitudinal location of the target seas, it was relevant to use an equal-area polar projection. Therefore, the NASA 'Ease-Grid' was employed. The system of coordinates of the WGS-84 (World Geodetic System 1984) is at the basis of 'Ease-Grid'.

2. Correction of Automatic Cloud Masking in the images from SeaWiFS in 1998–2001. In all images of the OC CCI product obtained in 1998–2001 (when only the SeaWiFS sensor wasoperational), all putative bloom areas proved to be masked. The errors of automatic cloud masking most probably resulted from very high values of brightness stemming from bloom areas (comparable with cloud-produced signals), which may have led to possible mistakes in the masking algorithm. The problem was overcome via manual processing of the data of a lower level, i.e. directly from the SeaWiFS level 2 product (http://oceancolor.gsfc.nasa.gov/cgi/browse.pl?sen=am) for the period of 1998–2001 in all studied areas. As a result, in the RGB-images the areas masked as clouds in OC CCI images proved to exhibit large bloom areas with the brightness of signals typical of *E. huxleyi*. This approach was legitimate as OC CCI data obtained by different sensors have been brought to the SeaWiFS standard channels, and the entire data time series (1998-2016) was radiometrically uniform.

3. Filling Missing Pixels Masked as Ragged Clouds. In the case of ragged clouds, some pixels of RGB images are not informative. A special algorithm for filling such gaps included averaging of $R_{rs}(\lambda)$ values from neighboring pixels and from temporarily previous and following images of the same pixel. The use of this algorithm in each of the cloud-masked images of the areas studied over 19 years and included in the OC CCI product helped increase the analysed area, sometimes to a significant extent. Calculated from 1998 to 2016 as arithmetic means for the Barents, Bering, North, Norwegian and Greenland seas, the quantitative estimates of such an increase attained for each 8-day-averaged image reached, respectively, ~107, 370, 31, 15, and 13 times. Thus, obtained were images with significantly larger cloud-free areas assuring a more accurate estimation of the borders of bloom areas, and their displacement, as well as of bloom areas per se.

Examples of products visualizations (for the North Sea) are shown in Figure 2.

## 3 Data sources

Data on $R_{rs}$ in six channels (centered at 412, 443, 490, 510, and 670 nm) are from the OC CCI product (Ocean Colour Climate Change Initiative dataset, Version 3.0, European Space Agency, available online at http:// www.esa-oceancolour-cci.org/).

For the bio-optical retrieval algorithm validation, we employed the PANGAEA database (www.pangaea.de) of the concentration of coccoliths within the target coccolithophore blooms in the North Atlantic including the North and Norwegian Seas (Charalampopoulou et al. 2008, 2011).

The bio-optical in situ database spanning between 1997 and 2012 (16 years) was employed for ocean-colour satellite applications as having a global coverage (Valente et al., 2016). The data were acquired from several sources: MOBY



(Marine Optical Buoy), BOUSSOLE (BOUée pour l'acquiSition d'une Série Optique à Long termE), AERONET-OC (Aerosol Robotic NETwork-Ocean Color), SeaBASS (SeaWiFS Bio-optical Archive and Storage System), NOMAD (NASA bio-Optical Marine Algorithm Dataset), MERMAID (MERIS Match-up In situ Database), AMT (Atlantic Meridional Transect), ICES (International Council for the Exploration of the Sea), HOT (Hawaii Ocean Time-series), and GeP&CO

(Geochemistry, Phytoplankton, and Color of the Ocean). This database comprises a large number of variables, including the spectral remote sensing reflectance, $R_{rs}$, and chlorophyll-a concentration.

Data on mixed layer depth (MLD) were derived from the Montegut climatology (Montegut et al. 2004).

Data on bathymetry inherent in the target seas were taken from the website http://www.ngdc.noaa.gov/mgg/bathymetry/arctic/arctic.html (Jakobsson et al. 2012).

The GLobal Ocean Data Analysis Project (GLODAP) database (Key et al., 2015; Olsen et al., 2016), http://cdiac. ornl.gov/oceans/GLODAPv2/ was employed for pairing in situ $NO_3$ values at those points for which in situ $pCO_2$ values were available. In the cases when the desired $NO_3$ matching values were unavailable in the GLODAP database, the respective data were employed from the World Ocean Atlas 2013 (WOA13, NOAA, Garcia et al., 2014; https://www.nodc.noaa.gov/OC5/woa13/).

The SOCAT v4 database (The Surface Ocean $CO_2$ Atlas, Bakker et al., 2016; http://www.socat.info/access.html) comprises more than 6 million $pCO_2$ measurements performed at latitudes north of 40°N. The data employed by us from SOCAT V4 database met the following requirements: (*1*) measurements conducted during 1998–2016 and within a 10 m top layer (if there were data from several depths, the measurements from the shallowest depth were used); (*2*) $pCO_2$ data should necessarily have both corresponding seawater salinity data and valid $R_{rs}$ spectra; (*3*) a daily mean $pCO_2$ value was employed

provided there were several in situ measurements; (*4*) $pCO_2$ measurements conducted at a distance not less than 8 km offshore (to avoid the impact of adjacency effect on $R_{rs}$ satellite data); (*5*) $pCO_2$ measurements were within the location and timing of *E. huxleyi* blooming; and (*6*) data used from SOCAT v4 database overlap the data from either the GLODAP database or the WOA13 climatology database (depending upon which one was used for comparison).

The GLobal Ocean Data Analysis Project (GLODAP) database (Key et al., 2015; Olsen et al., 2016), http://cdiac.

ornl.gov/oceans/GLODAPv2/ was employed for pairing in situ $NO_3$ values at those points for which in situ $pCO_2$ values were available. In the cases when the desired NO3 matching values were unavailable in the GLODAP database we resorted to the respective data from the World Ocean Atlas 2013 (WOA13, NOAA, Garcia et al., 2014; https://www.nodc.noaa.gov/OC5/woa13/).

## 4 Data spatio-temporal domain

The published dataset covers a time period of 19 years, from 1998 to 2016, with a time resolution of 8 days (a total of 874 time periods), and a spatial domain with the total area of 1,105,6800 km$^2$ at a resolution of 4x4 km, divided into 4 regions described in Table 1 and shown in Figure 3.




All data a represented in the Lambert Azimuthal Equal area projection with the parameters corresponding to the widespread NSIDC EASE-Grid North (EPSG: 3973) coordinate system.

The selection of 4 regions in this work resides in several reasons. They include all seas where coccolithophore blooms usually occur in subpolar and polar regions of the Northern Hemisphere (North, Norwegian, Greenland, Barents, Bering and

5    Labrador seas). The exclusion from our dataset of blooms occurring in the northern parts of Atlantic Ocean (see, e.g. Holligan et al. 1993) was dictated by some technical restrictions: the hydro-optical model employed for obtaining coccolith concentration values was based prevalently on the data from high-latitude areas, and thus should be at first validated for geographically different marine environments such as open parts of the Atlantic Ocean.

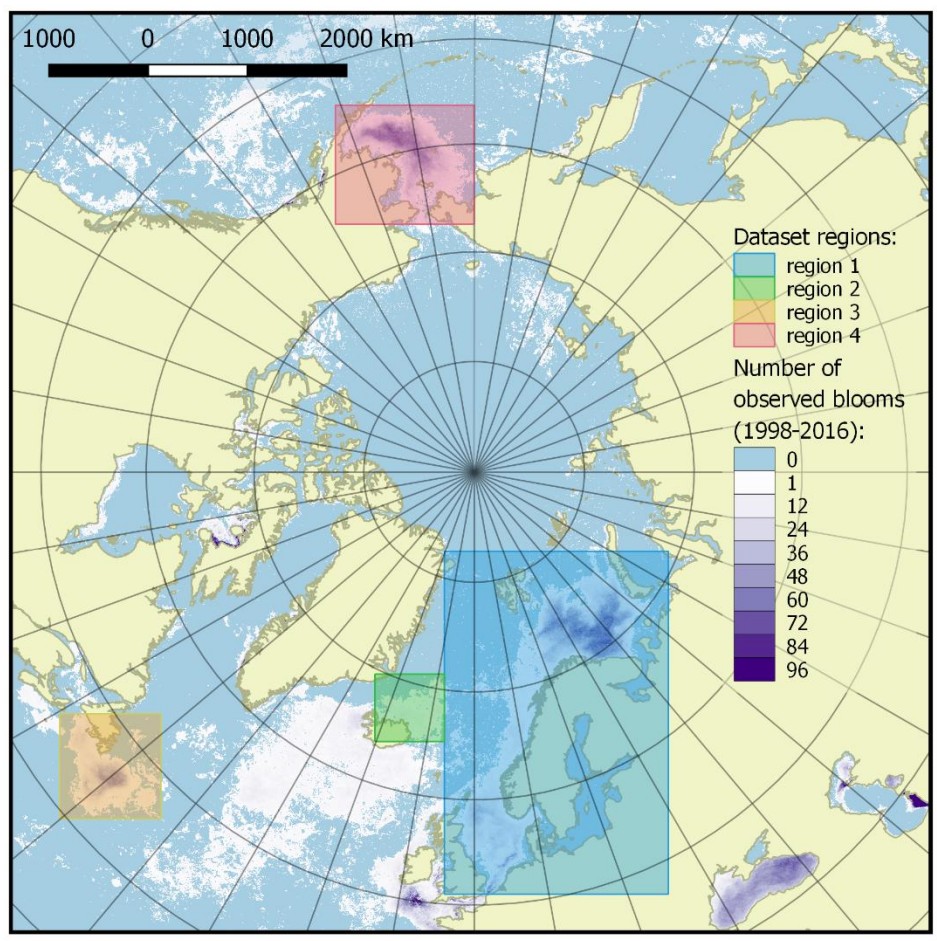

**Figure 3: Dataset of target spatial regions. Regions are shown as coloured boxes, and the colourbar indicates the number of bloom observations in each pixel over the time period 1998 - 2016.**



## 5 Dataset overview

The 19-year period data covers 4 blooming regions differing in nature. This allows to evaluate the bloom-related processes at different scales and time intervals in order to reveal both interannual dynamics and seasonal variations of parameters relevant to the bloom phenomenon. *E. huxleyi* blooms in the Arctic and Subarctic seas are characterized by significant

5    instability: the difference in intensity of blooming in different years can reach tens of times. Figure 4 and Table 2 collectively illustrate for the above four marine regions the temporal dynamics in bloom intensity (i.e. blooming area). For example, in the Bering Sea (region 4), the most extensive blooms were observed exclusively from 1998 to 2001, but later on, their intensity decreased drastically. In region 1, mainly in the Barents, Norwegian and Northseas, the blooming activity over the years we are reporting on was very irregular, with a peak in 2016.

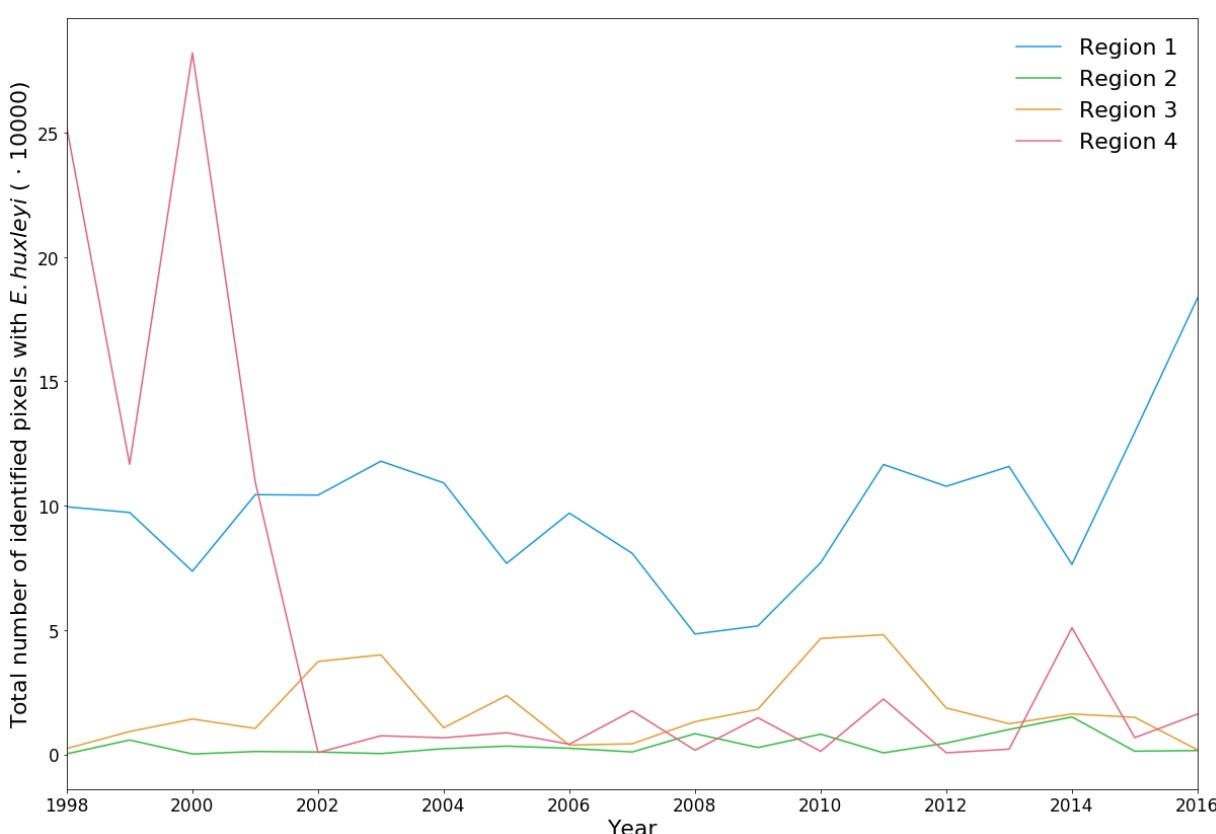

**Figure 4: Total number dynamics of identified pixels with *E. huxleyi* for each blooming season in the period 1998-2016 within the four regions specified in Figure 3.**

With the data collected, it's possible to highlight the patterns of development of the regularly occurring blooms. They can be characterized with the beginning/end of blooming periods, and the overall dynamics of coccolith concentration during the

15    blooms. Such patterns can be established based on the published dataset. Figure 5 shows an example of bloom development in the Greenland Sea (region 2) in the period June 26 - August 13, 2014. However, these periods are generally unstable,



which is clearly seen in Figure 6, which displays the blooming area configuration in July, 20 for different years for the same area.

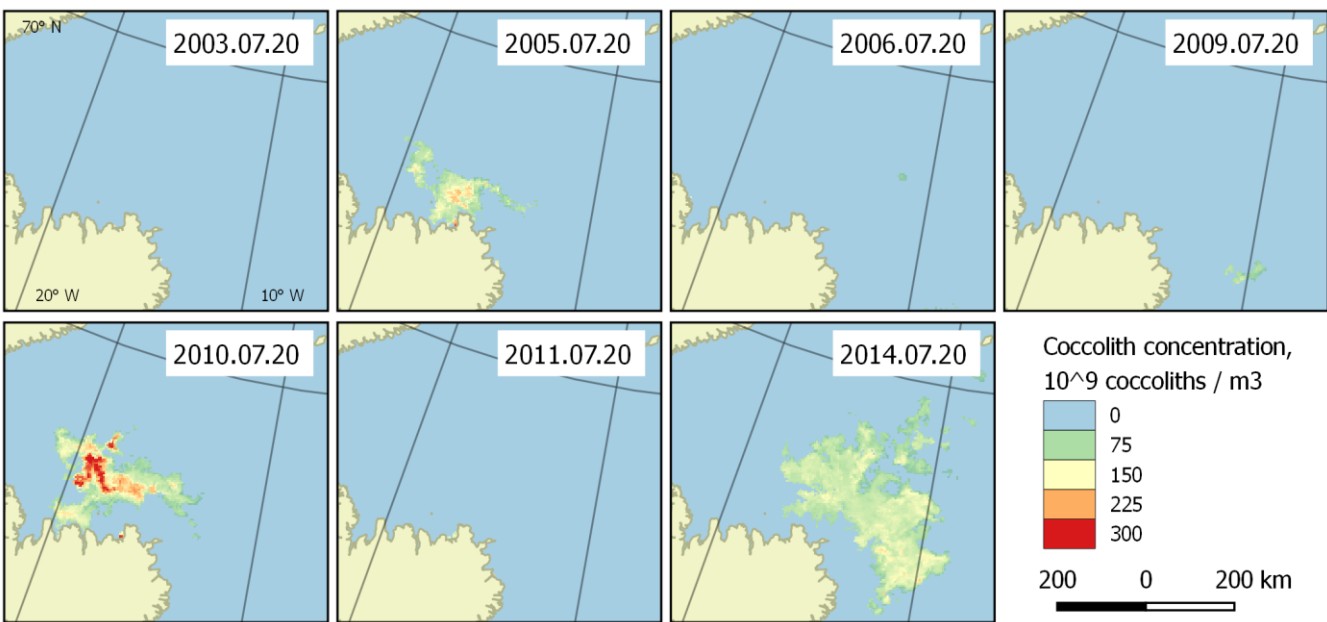

**Figure 5: Bloom development in the Greenland Sea (region 2) in June-August 2014. The peak falls on July 20.**

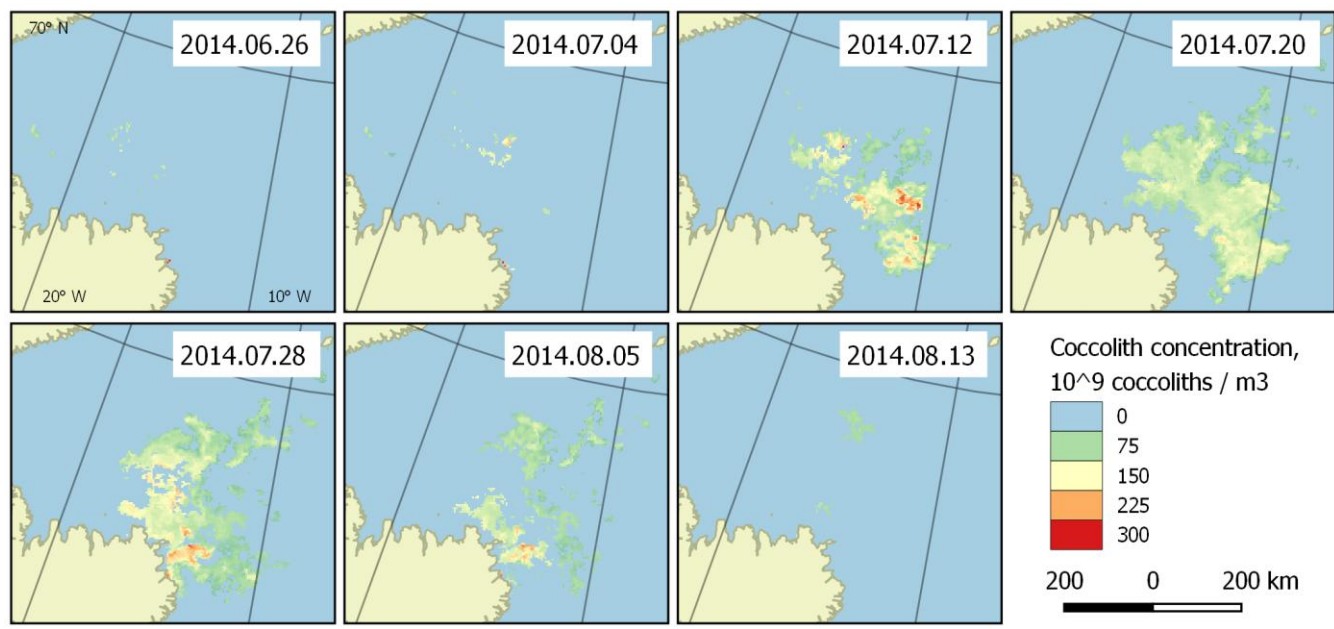

**Figure 6: Bloom intensity in the Greenland Sea (region 2) on July 20 in different years. Its instability is obvious.**



Technically, each dataset contains 4 subdatasets: bloom status, coccolith concentration, particulate organic carbon content and $CO_2$ partial pressure in water driven by coccolithophores. The last three categories contain the parameter values directly calculated. The first subdataset contains information about the quality and content of data. This information is organised as a set of flags attributed to data on reliable observations of blooming presence or absence, or inaccurate data (usually due to clouds) as well as data on coastal land. Figure 7 provides both an example of a status matrix and the matrix containing coccolith concentration values.

Region 1
2011.08.05 - 2011.08.12

bloom_status dataset content

- ▦ Trusted observation - no bloom
- ▦ Trusted observation - bloom
- ☐ Cloud / bad observation
- ☐ Land

100 0 100 200 300 400 km

Region 1
2011.08.05 - 2011.08.12

coccolith_concentration dataset
content

Coccolith concentration, 10^9 coccoliths / m3
- ☐ No data
- ■ ~0
- ■ 200
- ■ 400
- ■ 600
- ■ 800

100 0 100 200 300 400 km

**Figure 7: Dataset content example (region 1, 2011.08.05). a - bloom status subdataset visualization, b - coccolith concentration subdataset visualization.**

## 6 Data availability

Dataset is available on Zenodo (Kondrik et al. 2018b; https://doi.org/10.5281/zenodo.1402033). Data granules are divided into directories by regions and years, each child directory contains files with 8-day periods data on the bloom status, coccolith concentration, PIC, $\Delta p CO_2$. Data are stored in NetCDF4 format with GDAL-support, that allows to use the data

immediately with any NetCDF-based or GIS software. Tips about how to read the data and QGIS styles for fast visualizations are also provided.

## 7 Conclusions

We have composed a detailed 19-year dataset of *E. huxleyi* blooms in the Arctic and Subarctic seas, including the information about their influence on the carbon cycle in the ocean. These data are based mostly on satellite remote sensing observations, but also on available shipborne measurements and results of processing with authorial algorithms. We hope that the publication of these data, on the one hand, will promote further studies aimed at elucidating *E. huxleyi* bloom driving mechanisms and their forcing factors and, on the other hand, will facilitate understanding the patterns of this phenomenon distribution and its impact on the ocean and the atmosphere.

## Author contributions

Dmitry Pozdnyakov is responsible for theoretical background and methodology development. Dmitry Kondrik also contributed to theoretical background research, and responsible for data processing algorithms development and programming. Eduard Kazakov conceived the dataset structure and contributed to data processing algorithms programming, data analysis and visualizations. All authors equally contributed to the writing of the manuscript and data quality control.

## Competing interests

The authors declare that they have no competing interests.

## Acknowledgments

The Surface Ocean $CO_2$ Atlas (SOCAT) is an international effort, endorsed by the International Ocean Carbon Coordination Project (IOCCP), the Surface Ocean Lower Atmosphere Study (SOLAS) and the Integrated Marine Biosphere Research (IMBeR) program, to deliver a uniformly quality-controlled surface ocean $CO_2$ database. The many researchers and funding agencies responsible for the collection of data and quality control are thanked for their contributions to SOCAT.
We express our particular gratitude for the financial support of this study provided by the Russian Science Foundation (RSF) under the project 17-17-01117.



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



**Table 1.** Spatial regions description

| Region number | Extent coordinates (NSIDC EASE-Grid North, EPSG:3973) | | Region Area, km$^2$ | Contained waters |
|---|---|---|---|---|
| 1 | Xmin | -300000.00 | 7 819 600 | The Barents, Norwegian, North seas, the Northern part of the Greenland Sea |
| | Ymin | -4260000.00 | | |
| | Xmax | 1960000.00 | | |
| | Ymax | -800000.00 | | |
| 2 | Xmin | -1000000.00 | 476 000 | Southern part of the Greenland sea, Western part of the Norwegian Sea |
| | Ymin | -2720000.00 | | |
| | Xmax | -300000.00 | | |
| | Ymax | -2040000.00 | | |
| 3 | Xmin | -4180000.00 | 1 081 200 | Southern part of the Labrador Sea, the North Atlantic Ocean part to the south of the Labrador Sea |
| | Ymin | -3500000.00 | | |
| | Xmax | -3160000.00 | | |
| | Ymax | -2440000.00 | | |
| 4 | Xmin | -1400000.00 | 1 680 000 | The Bering Sea |
| | Ymin | 2500000.00 | | |
| | Xmax | 0.00 | | |
| | Ymax | 3700000.00 | | |



**Table 2.** Total number of identified pixels with *E. huxleyi* for each blooming season in the period 1998-2016 within the four regions.

| Year | Total number of pixels with *E. huxleyi* | | | | Year | Total number of pixels with *E. huxleyi* | | | |
|------|----------|----------|----------|----------|------|----------|----------|----------|----------|
|      | Region 1 | Region 2 | Region 3 | Region 4 |      | Region 1 | Region 2 | Region 3 | Region 4 |
| 1998 | 99538    | 214      | 2336     | 252003   | 2008 | 48399    | 8319     | 13131    | 1656     |
| 1999 | 97259    | 5754     | 9168     | 116622   | 2009 | 51620    | 2745     | 18102    | 14749    |
| 2000 | 73642    | 138      | 14205    | 282046   | 2010 | 77050    | 8110     | 46591    | 1232     |
| 2001 | 104425   | 1142     | 10432    | 109541   | 2011 | 116555   | 603      | 48101    | 22259    |
| 2002 | 104237   | 949      | 37335    | 694      | 2012 | 107791   | 4532     | 18630    | 618      |
| 2003 | 117877   | 312      | 40018    | 7466     | 2013 | 115764   | 10011    | 12302    | 2079     |
| 2004 | 109156   | 2275     | 10686    | 6657     | 2014 | 76396    | 15047    | 16245    | 50900    |
| 2005 | 76768    | 3300     | 23651    | 8679     | 2016 | 129569   | 1265     | 14890    | 6705     |
| 2006 | 97004    | 2444     | 3729     | 4061     | 2017 | 183546   | 1536     | 1779     | 16184    |
| 2007 | 80835    | 955      | 4237     | 17505    |      |          |          |          |          |