# Peer review of "A synthetic satellite dataset of *Emiliania huxleyi* blooms spatiotemporal distributions and their impacts on Arctic and Subarctic marine environments (1998-2016)"

_Earth System Science Data, 2018_

## Referee Comment (RC1) · A.J. Poulton (Referee) · 5 Nov 2018

In this study, Kondrik et al. have compiled satellite observations of coccolithophore blooms in the high-latitude northern hemisphere and combined them with various algorithms, published by the authors, to estimate coccolith concentrations and the impact of coccolithophores on the air-sea CO2 fluxes. The dataset is of considerable interest, with coccolithophore blooms in the high-latitude polar seas generally understudied and often poorly sampled in situ. The 18-year time-series of observations represents an exciting opportunity to examine temporal trends over a relatively long period and I am

sure the dataset will be used extensively. The manuscript is well written and I only have minor comments/suggestions for further clarity.

pg 1, Ln 1 - How do you know its E. huxleyi rather than other coccolithophores? Would it not be safer to say coccolithophores? Though many factors make E. huxleyi the most likely source of satellite-detectable reflectance, other coccolithophores can bloom and some can be a significant presence within blooms. Also, its more typical to give the full species name (i.e. Emiliania huxleyi).

pg 1, Ln 7 – What do the authors mean by 'activity' in the context of the first line of the paragraph? Distribution and impact on the air-sea flux of $CO_2$ is what is presented.

pg 1, Ln 16 – 'Ongoing climate change is a background of numerous emerging hot topics' seems a rather cryptic opening line for the paper and it's not obviously clear what the authors mean.

pg 1, Ln 20 – 'most widespread in the world's oceans': please clarify this statement, do you mean 'the' most widespread coccolithophore?

pg 1, Ln 25 – Rivero-Calle et al. (2015) show increases in occurrence across the North Atlantic rather than a polewards expansion. Other authors have discussed polar expansion ranges (e.g. Smyth et al., 2004; Winter et al., 2014) and are more relevant to the current study.

pg 2, Ln 6 – Please rephrase 'solely satellite remote sensing approach means..'.

pg 2, Ln 21 – Please explain 'viz. North', do you mean the North Atlantic?

pg 4, Ln 30-32 – Please note that the use of a fixed carbon mass per coccolith (m) has limitations and that coccolith content between different morphotypes of E. huxleyi can be considerable (e.g., Poulton et al., 2011; Müller et al., 2015) and may lead to over- or underestimation depending on which morphotype(s) is present in the bloom. This directly influences the scaling up of coccolith mass to PIC content in this study, and is an important factor when considering bloom PIC production (see e.g. Poulton et al.,

2013; Balch et al., 2014).

Figure 2c - What are the units for panel c? Tons per unit area/pixel? Would it not make more sense to express in similar volumetric units as in panel b (i.e. m-3)? It is also not clear how the authors get to 30 tons of PIC; e.g. 250-400 x109 coccoliths m-3 equates to ∼50 to 80 mg C m-3 or ∼4 to 7 mmol C m-3 based on a coccolith mass of 0.2 pg C.

References

Balch, W.M., Drapeau, D.T., Bowler, B.C., Lyczkowski, E.R., Lubelczyk, L.C., Painter, S.C., and Poulton, A.J.: Surface biological, chemical, and optical properties of the Patagonian Shelf coccolithophore bloom, the brightest waters of the Great Calcite Belt, Limnol. Oceanogr., 59, 1715-1732, doi: 10.4319/lo.2014.59.5.1715.

Müller, M.N., Trull, T.W., and Hallegraeff, G.M.: Differing responses of three Southern Ocean Emiliania huxleyi ecotypes to changing seawater carbonate chemistry, Mar. Ecol. Prog. Ser., 531, 81-90, doi: 10.3354/meps11309, 2015.

Poulton, A.J., Young, J.R., Bates, and Balch, W.M.: Biometry of detached Emiliania huxleyi coccoliths along the Patagonian Shelf, Mar. Ecol. Prog. Ser., 443, 1-17, doi: 10.3354/meps09445, 2011.

Poulton, A.J., Painter, S.C., Young, J.R., Bates, N.R., Bowler, B., Drapeau, D., Lyczsckowski, E., and Balch, W.M.: The 2008 Emiliania huxleyi bloom along the Patagonian Shelf: Ecology, biogeochemistry, and cellular calcification, Global Biogeochem. Cycl., 27, 1-11, doi: 10.1002/2013GB004641, 2013.

Smyth, T.J., Tyrrell, T., and Tarrant, B.: Time series of coccolithophore activity in the Barents Sea, from twenty years of satellite imagery, Geophys. Res. Lett., 31, L11302, doi: 10.1029/2004GL019735, 2004.

Winter, A., Hendericks, J., Beaufort, L., Rickaby, R.E.M., and Brown, C.W.: Poleward expansion of the coccolithophore Emiliania huxleyi, J. Plankton Res., 36, 316-325, doi: 10.1093/plankt/fbt110, 2014.

---

## Author Comment (AC1) · 7 Nov 2018

Dear Dr. Poulton,

Thank you for your thoughtful comments and recommendations. We are especially appreciative of the list of references.

Below are our answers.

Pg.1. Ln. 1: a) We will certainly change E. huxleyi for Emiliania huxleyi.

[Figure]

b) For all target seas we collected published reports from in situ/shipborne/laboratory studies explicitly indicating that the coccolithophore blooms were produced by E. huxleyi (see the attached specific list of references) with two exceptions for the Norwegian and Iceland seas, where along with E. huxleyi, Coccolithus pelagicus composes the coccolithophore community. However, as in situ determinations showed in the overwhelming cases the concentrations of cells of Coccolithus pelagicus were marginal (see e.g. Dylmer et al., 2015). This is the reason why we prefer leaving E. huxleyi instead of coccolithophores. A large number of papers on calcifying alga blooms in our targeted seas define the bloom-producing species as E. huxleyi.

Pg. 1, Ln. 7: By "activity" we meant the release of $CaCO_3$ in water and a decrease of uptake of dissolved $CO_2$ by E. huxleyi cells (e.g. Kondrik et al., 2018). In the revised version of the paper we will specify the actual meaning of the employed word "activity".

Pg. 1., Ln.16: It appeared to us that the issue of consequences of ongoing climate change–driven consequences is presently a commonplace, not requiring any further specialization. Indeed, the consequences are multifaceted, with numerous forward and feedback interactions and relate to many spheres of knowledge. So we choose to extend this phrase a little bit and provide this sentence with a reference that reasonably overarches the main dimensions of this phenomenon.

Pg. 1, Ln. 20: Yes, we will change for "the most widespread coccolithophore".

Pg. 1. Ln. 25; You are right, and we will add the reference "Winter et al., 2014".

Pg. 2., Ln 6:We agree that this phrase is kind of awkward and we will reword it as follows: "solely satellite remote sensing approach is. . ."

Pg. 2. Ln. 21: the following change will be made: the North, Labrador (with adjacent North Atlantic open waters), Norwegian, Barents, Greenland and Bering seas.

Pg. 4, Lns 30-32+ Figure 2c: The total content of PIC, Mpic, was determined for each 8-day time-period through multiplication of the carbon mass per coccolith, m, the coccolith concentration, Ccc, MLD and the bloom area, S. The value of m was equalled to 0.2pg. While most historical reports support this estimation, it is likely that the employment of this conversion might lead to either (i) some underestimation of PIC since it nevertheless neglects rare, relatively large, suspended calcite particles (PIC concentration per coccolith is ∼0.26 pg by Balch et al.(1991) and 0.5-0.6 pg by Holliganet al.(1983)) or (ii) some underestimation as there are in situ data indicating that many coccoliths in E. huxleyi blooms are either fragmented due to wave action (Holliganet al. 1993b) or just of a smaller size (PIC concentration is 0.13 pg) (Fernandez et al. 1993, Fritz 1999). Thus on balance, the selected value of m, in all probability, is a reasonably good estimate which is supported by the historical literature (Balch et al. 2005). The respective details are provided in section 2. Accordingly, the numbers in Figure 2c are indeed in tons as they reflect the content of PIC in a pixel-size column with the vertical extent equal to the respective MLD that was ascribed to each pixel within the bloom area. The respective methodology is described in detail in Kondrik et al., 2017 and will be given in the text.

Again, we express our gratitude to the referee for his very valuable comments.

Publications explicitly indicating the kind of coccolithophore species forming bloom in the target seas:

Barents Sea (Olson & Strom, 2002)

Bering Sea (Sukhanova and Flint, 1998)

North Sea (Holligan et al., 1993b; Buitenhuis et al., 1996)

Norwegian Sea (Baumann et al., 2000)

Labrador Sea (Okada & McIntyre, 1979)

North Atlantic (Holligan et al., 1993a)

Greenland Sea (Dylmer et al., 2015)
References

Balch, W. M., P. M. Holligan, S. G. Ackleson, and K. J. Voss. 1991. "Biological and Optical Properties of Mesoscale Coccolithophore Blooms in the Gulf of Maine." Limnology and Oceanography 36: 629–643. doi:10.4319/lo.1991.36.4.0629

Balch, W. M., H. Gordon, B. C. Bowler, D. T. Drapeau, and E. S. Booth. 2005. "Calcium Carbonate Measurements in the Surface Global Ocean Based on Moderate-Resolution Imaging Spectrometer Data." Journal of Geophysical Research110: C07001. doi:10.1029/2004JC002560

Baumann, Karl-Heinz &Andruleit, HA &Samtleben, C. (2000). Coccolithophores in the Nordic Seas: Comparison of living communities with surface sediment assemblages. Deep-sea Research Part Ii-topical Studies in Oceanography - DEEP-SEA RES PT II-TOP ST OCE. 47. 1743-1772. 10.1016/S0967-0645(00)00005-9.

Buitenhuis, E., J. van Bleijswijk, D. Bakker, and M. Veldhuis, Trends in inorganic and organic carbon in a bloom of Emilianiahuxleyi in the North Sea, Mar. Ecol. Prog. Ser., 143, 271-282, 1996

Dylmer, C. V., Giraudeau, J., Hanquiez, V., &Husum, K. (2015). The coccolithophoresEmilianiahuxleyi and Coccolithuspelagicus : Extant populations from the Norwegian–Iceland Seas and Fram Strait. Deep Sea Research Part I: Oceanographic Research Papers. https://doi.org/10.1016/j.dsr.2014.11.012

Fernandez, E., P. Boyd, P. M. Holligan, and D. S. Harbour. 1993. "Production of Organic and Inorganic Carbon within a Large Scale Coccolithophore Bloom in the North Atlantic Ocean." Marine Ecology Progress Series 97: 271–285. doi:10.3354/meps097271

Fritz, J. J. 1999. "Carbon Fixation and Coccolith Detachment in the Coccolithophore E. Huxleyi in Nitrate-Limited Cyclostats." Marine Biology 133: 509–518. doi:10.1007/s002270050491

Holligan, P. M., M. Viollier, C. Dupouy, and J. Aiken. 1983. "Satellite Studies on the

Distribution of Chlorophyll and Dinoflagellate Blooms in the Western English Channel." Continental Shelf Research 2 (2–3): 81–96. doi:10.1016/0278-4343(83)90009-2

Holligan, P. M., E. Fernández, J. Aiken, W. M. Balch, P. Boyd, P. H. Burkill, M. Finch, et al. 1993a. "A Biogeochemical Study of the Coccolithophore, Emilianiahuxleyi, in the North Atlantic." Global Biogeochemical Cycles 7: 879–900. doi:10.1029/93GB01731.

Holligan PM, Groom SB, Harbour DS (1993b) What controls the distribution of the coccolithophore, Emilianiahuxleyi, in the North Sea? Fish Oceanogr 2(3/4):17

Kondrik Dmitry, Dmitry Pozdnyakov & Lasse Pettersson (2017) Particulate inorganic carbon production within E. huxleyi blooms in subpolar and polar seas: a satellite time series study (1998–2013), International Journal of Remote Sensing, 38:22, 6179-6205, DOI: 10.1080/01431161.2017.1350304

Kondrik, D. V., Pozdnyakov, D. V., and Johannessen, O. M.: Satellite evidence that E. huxleyi phytoplankton blooms weaken marine carbon sinks, Geophysical Research Letters, 45, 2, 846-854, doi:10.1002/2017GL076240, 2018

Okada, H., and A. McIntyre. 1979. "Seasonal Distribution of Modern Coccolithophores in the Western North Atlantic Ocean." Marine Biology 54: 319–328. doi:10.1007/BF00395438

Olson, M. B., and S. L. Strom. 2002. Phytoplankton growth, microzooplankton herbivory and community structure in the southeast Bering Sea: Insight into the formation and temporal persistence of an Emilianiahuxleyi bloom. Deep-Sea Res. II 49: 5969–5990

Sukhanova, I. N., and M. V. Flint, Anomalous blooming of coccolithophorids over the eastern Bering Sea shelf, Oceanology, 38, 502 – 505, 1998

Winter, A., Hendericks, J., Beaufort, L., Rickaby, R.E.M., and Brown, C.W.: Poleward expansion of the coccolithophore Emiliania huxleyi, J. Plankton Res., 36, 316-325, doi: 10.1093/plankt/fbt110, 2014

---

## Referee Comment (RC2) · NEUKERMANS (Referee) · 29 Nov 2018

Kondrik and collaborators present a 19-year satellite time series of Emiliania huxleyi bloom area, calcite content, and associated increase in in-water pCO2 in four selected areas of the high-latitude northern hemisphere. The dataset is only partly unique, in the sense that a 19-year global remote sensing dataset of E. huxleyi bloom extent, coccolith concentration, and PIC content can also be easily obtained elsewhere. Therefore uniqueness only applies to pCO2. This dataset could be useful, but I request a few substantial modifications that I believe are necessary to improve understanding and

quality of the dataset: (1) some flaws in the dataset (pointed out below, 1a and 1b) will need to be fixed, (2) error estimates for remotely sensed quantities must be provided, and (3) in its present form, the study/data is not correctly positioned within the state-of-the-art literature and other available datasets.

(1a) It appears from Fig. 4 that the E. huxleyi bloom dataset includes false positives, a problem that is particularly evident in the Bering Sea (1998-2001) where the authors have detected blooms initiating in winter and lasting about 10 months as previously reported from ocean colour remote sensing data (Iida et al., 2002). However, ship-borne measurements have identified resuspended diatom frustules as the cause of these bright waters in winter-spring instead of E. huxleyi blooms (Broerse et al., 2003). This invalidates the authorial E. huxleyi bloom detection algorithm and all derived products in the Bering Sea from late fall to spring. I further fail to see how the algorithms used by the authors (Kondrik et al. 2017; Kondrik et al. 2018) to detect E. huxleyi blooms present an advance to NASA's standard method of E. huxleyi bloom classification (Brown and Yoder, 1994), and many other subsequent bloom detection methods (Iglesias-Rodriguez et al., 2002; Iida et al., 2002; Iida et al., 2012; Moore et al., 2012). (1b) The remote sensing algorithm for pCO2 estimation is a simple linear regression between observations of Delta_pCO2 and remote sensing reflectance Rrs in a blue waveband. This relationship is strictly empirical and does not appear to have theoretical grounds; I believe the user should be aware of this. Not surprisingly, there is an enormous spread along this regression line such that for a given reflectance value the estimated Delta_pCO2 has a confidence interval with a width of 50 ppm and even wider for denser blooms. Furthermore, the residuals of the regression are clearly unevenly distributed, with a strong tendency to underestimate Delta_pCO2 at higher reflectances. This relationship should be explicitly stated, which is not presently the case, including all relevant regression statistics, and especially a figure showing the observations and the fitted line so that the user can better grasp the errors of the algorithm. (2) Whereas the statistics of the validation of the retrieved coccolith concentration are given in section 2.2, the accompanying figure is missing.
No uncertainty assessment is given for pCO2 (see previous comment). (3) A 19-year global remote sensing dataset of PIC concentration merging all ocean colour satellite missions can be obtained here: http://www.globcolour.info/ in temporal resolutions ranging from daily to monthly, spatial resolution ranging from 4km to 100km, and various geographical projections. From PIC concentration, coccolith concentration can be derived using a fixed mass per coccolith (as you do too), and PIC content can also be easily derived by combining with a climatology for Mixed layer depth available here http://www.ifremer.fr/cerweb/deboyer/mld/Surface_Mixed_Layer_Depth.php. I therefore suggest you remove all statements of uniqueness of your PIC dataset (e.g., page 2, lines 24-26). The statements on page 2 lines 11-16, "Prior to the publication of Kondrik et al. (2018), no attempts have been undertaken to retrieve from space... No concatenated time series data are available to date on the associated bloom intensity..." are thus simply incorrect. I also suggest you appropriately reference the work of (Shutler et al., 2013) entitled "Coccolithophore surface distributions in the North Atlantic and their modulation of the air-sea flux of CO2 from 10 years of satellite Earth observation data Âż, which is very similar to your work on remote sensing of pCO2 in Ehux blooms, but is mentioned nowhere. Page2 Line 8-10: "Until recently, only few satellite studies were published on the typical locations of E. huxleyi blooms and associated concentrations of PIC in surface waters within the bloom area". It appears to me you missed a vast body of literature: (Balch et al., 1991; Balch et al., 1996; Gordon et al., 2001; Smyth et al., 2004; Signorini and McClain, 2009; Moore et al., 2012; Hopkins et al., 2015; Balch et al., 2016; Neukermans et al., 2018) etc.

Further comments : Title : add "blooms" after "E. huxleyi" Abstract : delete "detailed information on E. huxleyi impacts within the bloom area on marine environments", as this suggests that you are detailing ecological impacts

P1, L16 : "Ongoing climate change is a background of numerous emerging hot topics." is a rather meaningless opening sentence. P1 L25 : Rivero-Calle is not the right reference for poleward expansion of coccolithophores, instead use (Winter et al., 2014;

Neukermans et al., 2018). "gradually propagating in the poleward direction" ; the pole-ward expansion is not gradual, as expansion rates exhibit stark jumps as demonstrated in (Neukermans et al., 2018). P2, L1-4 : a lot of statements for only one reference. P2, L23 : replace 1918-2016 by 1998-2016 P2, L20 : remove "original" P3 L1 : spell out OC CCI P6 L1 : "in the cause of satellite processing" ?, rephrase P7 L10-15 and L24-28 : same paragraph appears twice. P7 L31 :"1,105,6800 km2" commas are in the wrong place

References : Balch WM, Bates NR, Lam PJ, Twining BS, Rosengard SZ, Bowler BC, Drapeau DT, Garley R, Lubelczyk LC, Mitchell C, et al. 2016. Factors regulating the Great Calcite Belt in the Southern Ocean and its biogeochemical significance. Global Biogeochem Cycles 30(8): 1124–1144. Wiley-Blackwell. doi: 10.1002/2016GB005414 Balch WM, Holligan PM, Ackleson SG, Voss KJ. 1991. Biological and optical prop-erties of mesoscale coccolithophore blooms in the Gulf of Maine. Limnol Oceanogr 36(4): 629–643. doi: 10.4319/lo.1991.36.4.0629 Balch WM, Kilpatrick KA, Holligan P, Harbour D, Fernandez E. 1996. The 1991 coccolithophore bloom in the central North Atlantic. 2. Relating optics to coccolith concentration. Limnol Oceanogr 41(8): 1684–1696. doi: 10.4319/lo.1996.41.8.1684 Broerse AT., Tyrrell T, Young J., Poulton A., Merico A, Balch W., Miller P. 2003. The cause of bright waters in the Bering Sea in winter. Cont Shelf Res 23(16): 1579–1596. doi: 10.1016/j.csr.2003.07.001 Brown CW, Yoder JA. 1994. Coccolithophorid blooms in the global ocean. J Geophys Res 99(C4): 7467. doi: 10.1029/93JC02156 Gordon HR, Boynton GC, Balch WM, Groom SB, Har-bour DS, Smyth TJ. 2001. Retrieval of coccolithophore calcite concentration from Sea-WiFS Imagery. Geophys Res Lett 28(8): 1587–1590. doi: 10.1029/2000GL012025 Hopkins J, Henson SA, Painter SC, Tyrrell T, Poulton AJ. 2015. Phenological charac-teristics of global coccolithophore blooms. Global Biogeochem Cycles 29(2): 239–253. doi: 10.1002/2014GB004919 Iglesias-Rodriguez MD, Brown CW, Doney SC, Kleypas JA, Kolber D, Kolber Z, Hayes PK, Falkowski PG. 2002. Representing key phytoplankton functional groups in ocean carbon cycle models: Coccolithophorids. Global Biogeochem Cycles 16(4): 47-1-47–20. doi: 10.1029/2001GB001454 Iida T,

Mizobata K, Saitoh S-I. 2012. Interannual variability of coccolithophore Emiliania huxleyi blooms in response to changes in water column stability in the eastern Bering Sea. Cont Shelf Res 34: 7–17. Pergamon. doi: 10.1016/J.CSR.2011.11.007 Iida T, Saitoh SI, Miyamura T, Toratani M, Fukushima H, Shiga N. 2002. Temporal and spatial variability of coccolithophore blooms in the eastern Bering Sea, 1998-2001. Prog Oceanogr 55(1–2): 165–175. Pergamon. doi: 10.1016/S0079-6611(02)00076-9 Moore TS, Dowell MD, Franz BA. 2012. Detection of coccolithophore blooms in ocean color satellite imagery: A generalized approach for use with multiple sensors. Remote Sens Environ 117: 249–263. doi: 10.1016/j.rse.2011.10.001 Neukermans G, Oziel L, Babin M. 2018. Increased intrusion of warming Atlantic water leads to rapid expansion of temperate phytoplankton in the Arctic. Glob Chang Biol 24(6): 2545–2553. Wiley/Blackwell (10.1111). doi: 10.1111/gcb.14075 Shutler JD, Land PE, Brown CW, Findlay HS, Donlon CJ, Medland M, Snooke R, Blackford JC. 2013. Coccolithophore surface distributions in the North Atlantic and their modulation of the air-sea flux of CO2 from 10 years of satellite Earth observation data. Biogeosciences 10(4): 2699–2709. doi: 10.5194/bg-10-2699-2013 Signorini SR, McClain CR. 2009. Environmental factors controlling the Barents Sea spring-summer phytoplankton blooms. Geophys Res Lett 36(10): L10604. doi: 10.1029/2009GL037695 Smyth TJ, Tyrell T, Tarrant B. 2004. Time series of coccolithophore activity in the Barents Sea, from twenty years of satellite imagery. Geophys Res Lett 31(11): L11302. doi: 10.1029/2004GL019735 Winter A, Henderiks J, Beaufort L, Rickaby REM, Brown CW. 2014. Poleward expansion of the coccolithophore Emiliania huxleyi. J Plankton Res 36(2): 316–325. Oxford University Press. doi: 10.1093/plankt/fbt110

---

## Author Comment (AC2) · 24 Dec 2018

1. Regarding the status of our database.

With all respect for the reviewer, we can't agree with the reviewer's opinion that if any dataset(s) including the parameter(s) listed in our paper already exist(s) then our dataset can not be qualified as unique. The uniqueness of our dataset resides in that that

(A) it combines a spatially and temporarily collocated setof parameters (not solely e.g.

[Figure]

coccolith concentration)inherent in /related to the E. huxleyi blooms phenomenon in a number of polar and subpolar marine regions

(B)over the satellite measurement period of nearly 20 years (1998-2016), it is

(C) based on merged data from several satellites of the modern era (such as SeaWiFS, MODIS, MERIS, VIIRS), and

(D) designed specifically for the user convenience. Thus importantly, the user does not need to compose such a comprehensive database but use the already collected and user-friendly organized data source. Incidentally, this is explicitly corroborated by the reviewer himself/herself: even a spaceborne database on coccolith concentration per se is not available and needs to be retrieved from satellite datasets of PIC.

Summing up:

Given that our E-huxleyi-focused a ready-made database is yet unparalleled in terms of its combined areal+temporal coverage (6 seas in 3 oceans, 19 years, respectively), and the number of concatenated variables/parameters, we insist that, to date, it is veritably unique.

Other critical remarks relating to the issue of our database are commented on below.

2. Regarding the presence or absence of E. huxleyi blooms in the Bering Sea.

We considered this issue in detail in our work (Kondrik et al., 2017a), and it would obviously be improper to give here the respective entire excerpt from the above paper. In capsule:

A. Broerse et al.(2003) recognized that the area in which they took water samples, was on the very edge of the "bright patch". They write: "From the 7 February 2001 satellite image (Fig. 1(5)), it is not clear whether the sampling transect actually reached the edge of bright water patch". It is also worth pointing out that along with the diatom frustules,Broerse et al. also found coccoliths in their samples.

[Figure]

B. The ability of this alga to vegetate under conditions of very low levels of down-welling PAR irradiance is documented by Okada and McIntyre (1979): they have shown through their around-the-year shipborne measurements in the Labrador Sea at a latitudinal location (e.g. Station 'Bravo,' 56.5 ° N) similar to the location of the turquoise area in the Bering Sea that E. huxleyi cells indeed vegetated over a very long time period including not only summer time but also the winter period.

C. The appearance of turquoise areas in pelagic marine waters is a very strong argument in favor of attributing them to E. huxleyi blooms as no other hydrocoles possess such optical properties, which would render the truly turquoise color of their communities when observed from above. As Shutler et al. (2010) point out, this is a unique characteristic within phytoplankton species. Optically, diatom frustules are not identical to coccoliths. So that they would not produce the same remote sensing reflectance spectrum as coccoliths do.

An additional, albeit unnecessary argument: the phenomenon of huge blooms of E. huxleyi with extraordinarily high concentrations of coccoliths lasted only a few years and since 2001 have never re-occurred while diatoms blooms and associated release of frustules arethe annual event in the Bering Sea.

D. Finally, (although this argument is certainly optional, it only makes us additionally confident of our interpretation and robustness of our E. huxleyi bloom identification algorithm) we revealed the driving mechanism of the phenomenon of E. huxleyi blooms of exceptional intensity during 1998-2001, but this is the subject of a new paper, and we can't disclose it before its publication (expected in 2019).

In light of the above, the reviewer's assertion that our algorithm is invalidated because of the "false positives" in the Bering Sea could not be accepted.

3. Regarding the contested adequacy of our retrieval algorithms.

3a. On the advantage of our coccolith concentration retrieval algorithm.

We are not going to discuss here the advantages and disadvantages of E. huxleyi bloom detection algorithms suggested by other workers: it deserves a separate paper. Iida et al. (2002) have done it in detail with respect to e.g. the Brown and Yoder (1994) algorithm and pointed to some problems with it. Incidentally, Brown and Yoder themselves acknowledged certain limitationsof a world-wide application of their algorithm. Moore et al. (2012) commented on the feasibility of the algorithms in question developed by other teams that the reviewer specified in the his/her list of references.

The advantages of our algorithm were discussed in Kondrik et al. (2017a), and we hope that the reviewer does not expect us to dwell upon them. They can be epitomized as follows: our algorithm

(i) was developed on the basisof a nearly 20 year merged and skillfully harmonized OC CCI data provided by SeaWiFS, MODIS, MERIS, and VIIRS sensors;a comparative analysis of the OC CCI,GlobColour products, as well as the products from the MEaSUREs was conducted to prove the preference of the OC CCI data.

(ii) is based on extensive statistical analysis of satellite spectrometric [Rrs(lambda)] data fromsix marine environments specifically at high northern latitudes inthe North Atlantic, Arctic and North Pacific Oceans;

(iii) employsseveral criteria in conjunction, viz.: (a) location of maxima at the wavelengths typical of E. huxleyi bloom in Rrs spectra; (b) Rrs absolute value ranges at six wavelengths obtained through a dedicated/ large-size statistical sets of spaceborne data from the six seas; (c) consistency with the results of independent application of the BOREALI hydro-optical algorithm (Korosov et al., 2009; Kondrik et al., 2017a), which through retrieving inter alia the concentration of both coccoliths and chlorophyll-apermits to obtain the spatial distribution of the E. huxleyi bloom. This triple checking assured a higher reliability of the algorithm.

3b. Delta pCO2 retrieval algorithm

Again we believe that it would be improper to give here the respective entire excerpt from the paper on pCO2published in a refereed journal (Kondrik et al., 2018a). In a nutshell:

(i) the algorithm has the accuracy of delta pCO2 retrieval that is characterized by the following statistical parameters r2 = 0.54, pâĽł0.001, and RMSE = 23.4$\mu$atm;

(ii) the ensemble of blue data points in fig. 1 (Kondrik e al., 2018a) that looks like an "enormous spread" were obtained using climatological data and added solely to increase the statistical significance of the regression dependence established through using only in situ data that we could find for our study regions (red dots, their number is 187). Most of these points are within the declared error of 23.4 uatm; the indicated red points do not have the problem of Delta_pCO2 values overestimation indicated by the reviewer. It is also necessary to emphasize that a) "confidence interval" the reviewer refer tois in fact the "prediction limit" while the "confidence limit" has a much smaller variation (about 10 uatm). Also, it is important to be aware that the variation is given in uatm(units of partial pressure), but not in ppmas the reviewer writes.

(iii) all corrections for water temperature were duly conducted using the concurrently collected radiometric and IR satellite data.

(iv) the developed delta pCO2 regression dependence has a truly physical basis. Indeed, the increment of pCO2 in surface water within the E. huxleyi bloom is intimately related to the intracellular production of CO2 through the reaction of calcification and associated generation of coccoliths. The latter are very efficient reflectors of sun light coming into water (just because they don't absorb light but only reflect it). Therefore, the greater the amount of CO2 released through calcification, the more intense the optical signal coming out from the bloom area, especially at the wavelength of Rrs maximum – the parameter in our algorithm that is related to delta pCO2. Incidentally, returning to point 2C in our argumentations above, this is an important difference between coccoliths and diatomic frustules as the latter are not solely reflectors but also

absorbers.

4. The graphical illustration of validation of the retrievals of coccolith concentrations is available in our easily accessible papers published elsewhere, we doubt that the inclusion of those illustrations would be justified.

5. We acknowledge the reviewer's critical remarks in C3 –C4. All necessary changes are entered, the respective references [e.g. Shutler et al. (2010, 2013; Winter et al., 2014] are added to the reference list.

We certainly appreciate the list of references provided by the reviewer although, actually, we were aware of nearly all listed publications. The reason why they were not used is explained in point 1of our answers. As to the worksby Shutler et al. (2010,2013), it is indeed our flaw. We are earnestly grateful to the reviewer for this valuable critical remark.

---

## Author Response (AR2)

In this study, Kondrik et al. have compiled satellite observations of coccolithophore blooms in the high-latitude northern hemisphere and combined them with various algorithms, published by the authors, to estimate coccolith concentrations and the impact of coccolithophores on the air-sea $CO_2$ fluxes. The dataset is of considerable interest, with coccolithophore blooms in the high-latitude polar seas generally understudied and often poorly sampled in situ. The 18-year time-series of observations represents an exciting opportunity to examine temporal trends over a relatively long period and I am

sure the dataset will be used extensively. The manuscript is well written and I only have minor comments/suggestions for further clarity.

pg 1, Ln 1 - How do you know its E. huxleyi rather than other coccolithophores? Would it not be safer to say coccolithophores? Though many factors make E. huxleyi the most likely source of satellite-detectable reflectance, other coccolithophores can bloom and some can be a significant presence within blooms. Also, its more typical to give the full species name (i.e. Emiliania huxleyi).

pg 1, Ln 7 – What do the authors mean by 'activity' in the context of the first line of the paragraph? Distribution and impact on the air-sea flux of $CO_2$ is what is presented.

pg 1, Ln 16 – 'Ongoing climate change is a background of numerous emerging hot topics' seems a rather cryptic opening line for the paper and it's not obviously clear what the authors mean.

pg 1, Ln 20 – 'most widespread in the world's oceans': please clarify this statement, do you mean 'the' most widespread coccolithophore?

pg 1, Ln 25 – Rivero-Calle et al. (2015) show increases in occurrence across the North Atlantic rather than a polewards expansion. Other authors have discussed polar expansion ranges (e.g. Smyth et al., 2004; Winter et al., 2014) and are more relevant to the current study.

pg 2, Ln 6 – Please rephrase 'solely satellite remote sensing approach means..'.

pg 2, Ln 21 – Please explain 'viz. North', do you mean the North Atlantic?

pg 4, Ln 30-32 – Please note that the use of a fixed carbon mass per coccolith (m) has limitations and that coccolith content between different morphotypes of E. huxleyi can be considerable (e.g., Poulton et al., 2011; Müller et al., 2015) and may lead to over- or underestimation depending on which morphotype(s) is present in the bloom. This directly influences the scaling up of coccolith mass to PIC content in this study, and is an important factor when considering bloom PIC production (see e.g. Poulton et al.,

[Figure]

2013; Balch et al., 2014).

Figure 2c - What are the units for panel c? Tons per unit area/pixel? Would it not make more sense to express in similar volumetric units as in panel b (i.e. m-3)? It is also not clear how the authors get to 30 tons of PIC; e.g. 250-400 x109 coccoliths m-3 equates to ∼50 to 80 mg C m-3 or ∼4 to 7 mmol C m-3 based on a coccolith mass of 0.2 pg C.

References

Balch, W.M., Drapeau, D.T., Bowler, B.C., Lyczkowski, E.R., Lubelczyk, L.C., Painter, S.C., and Poulton, A.J.: Surface biological, chemical, and optical properties of the Patagonian Shelf coccolithophore bloom, the brightest waters of the Great Calcite Belt, Limnol. Oceanogr., 59, 1715-1732, doi: 10.4319/lo.2014.59.5.1715.

Müller, M.N., Trull, T.W., and Hallegraeff, G.M.: Differing responses of three Southern Ocean Emiliania huxleyi ecotypes to changing seawater carbonate chemistry, Mar. Ecol. Prog. Ser., 531, 81-90, doi: 10.3354/meps11309, 2015.

Poulton, A.J., Young, J.R., Bates, and Balch, W.M.: Biometry of detached Emiliania huxleyi coccoliths along the Patagonian Shelf, Mar. Ecol. Prog. Ser., 443, 1-17, doi: 10.3354/meps09445, 2011.

Poulton, A.J., Painter, S.C., Young, J.R., Bates, N.R., Bowler, B., Drapeau, D., Lyczsckowski, E., and Balch, W.M.: The 2008 Emiliania huxleyi bloom along the Patagonian Shelf: Ecology, biogeochemistry, and cellular calcification, Global Biogeochem. Cycl., 27, 1-11, doi: 10.1002/2013GB004641, 2013.

Smyth, T.J., Tyrrell, T., and Tarrant, B.: Time series of coccolithophore activity in the Barents Sea, from twenty years of satellite imagery, Geophys. Res. Lett., 31, L11302, doi: 10.1029/2004GL019735, 2004.

Winter, A., Hendericks, J., Beaufort, L., Rickaby, R.E.M., and Brown, C.W.: Poleward expansion of the coccolithophore Emiliania huxleyi, J. Plankton Res., 36, 316-325, doi: 10.1093/plankt/fbt110, 2014.

[Figure]

[Figure]

Earth Syst. Sci. Data Discuss.,
https://doi.org/10.5194/essd-2018-101-AC1, 2018
**Earth System** Open Access
**Science**
**Data** Discussions

Thank you for your thoughtful comments and recommendations. We are especially appreciative of the list of references.

Below are our answers.

Pg.1. Ln. 1: a) We will certainly change E. huxleyi for Emiliania huxleyi.

[Figure]

b) For all target seas we collected published reports from in situ/shipborne/laboratory studies explicitly indicating that the coccolithophore blooms were produced by E. hux­leyi (see the attached specific list of references) with two exceptions for the Norwegian and Iceland seas, where along with E. huxleyi, Coccolithus pelagicus composes the coccolithophore community. However, as in situ determinations showed in the over­whelming cases the concentrations of cells of Coccolithus pelagicus were marginal (see e.g. Dylmer et al., 2015). This is the reason why we prefer leaving E. huxleyi instead of coccolithophores. A large number of papers on calcifying alga blooms in our targeted seas define the bloom-producing species as E. huxleyi.

Pg. 1, Ln. 7: By "activity" we meant the release of CaCO3 in water and a decrease of uptake of dissolved CO2 by E. huxleyi cells (e.g. Kondrik et al., 2018). In the revised version of the paper we will specify the actual meaning of the employed word "activity".

Pg. 1., Ln.16: It appeared to us that the issue of consequences of ongoing climate change–driven consequences is presently a commonplace, not requiring any further specialization. Indeed, the consequences are multifaceted, with numerous forward and feedback interactions and relate to many spheres of knowledge. So we choose to extend this phrase a little bit and provide this sentence with a reference that reasonably overarches the main dimensions of this phenomenon.

Pg. 1, Ln. 20: Yes, we will change for "the most widespread coccolithophore".

Pg. 1. Ln. 25; You are right, and we will add the reference "Winter et al., 2014".

Pg. 2., Ln 6:We agree that this phrase is kind of awkward and we will reword it as follows: "solely satellite remote sensing approach is..."

Pg. 2. Ln. 21: the following change will be made: the North, Labrador (with adjacent North Atlantic open waters), Norwegian, Barents, Greenland and Bering seas.

Pg. 4, Lns 30-32+ Figure 2c: The total content of PIC, Mpic, was determined for each 8-day time-period through multiplication of the carbon mass per coccolith, m, the coc-

colith concentration, Ccc, MLD and the bloom area, S. The value of m was equalled to 0.2pg. While most historical reports support this estimation, it is likely that the employment of this conversion might lead to either (i) some underestimation of PIC since it nevertheless neglects rare, relatively large, suspended calcite particles (PIC concentration per coccolith is ∼0.26 pg by Balch et al.(1991) and 0.5-0.6 pg by Holliganet al.(1983)) or (ii) some underestimation as there are in situ data indicating that many coccoliths in E. huxleyi blooms are either fragmented due to wave action (Holliganet al. 1993b) or just of a smaller size (PIC concentration is 0.13 pg) (Fernandez et al. 1993, Fritz 1999). Thus on balance, the selected value of m, in all probability, is a reasonably good estimate which is supported by the historical literature (Balch et al. 2005). The respective details are provided in section 2. Accordingly, the numbers in Figure 2c are indeed in tons as they reflect the content of PIC in a pixel-size column with the vertical extent equal to the respective MLD that was ascribed to each pixel within the bloom area. The respective methodology is described in detail in Kondrik et al., 2017 and will be given in the text.

Again, we express our gratitude to the referee for his very valuable comments.

Publications explicitly indicating the kind of coccolithophore species forming bloom in the target seas:

Barents Sea (Olson & Strom, 2002)

Bering Sea (Sukhanova and Flint, 1998)

North Sea (Holligan et al., 1993b; Buitenhuis et al., 1996)

Norwegian Sea (Baumann et al., 2000)

Labrador Sea (Okada & McIntyre, 1979)

North Atlantic (Holligan et al., 1993a)

Greenland Sea (Dylmer et al., 2015)

[Figure]

(i) was developed on the basisof a nearly 20 year merged and skillfully harmonized OC CCI data provided by SeaWiFS, MODIS, MERIS, and VIIRS sensors;a comparative analysis of the OC CCI,GlobColour products, as well as the products from the MEaSUREs was conducted to prove the preference of the OC CCI data.

(ii) is based on extensive statistical analysis of satellite spectrometric [Rrs(lambda)] data fromsix marine environments specifically at high northern latitudes inthe North Atlantic, Arctic and North Pacific Oceans;

(iii) employsseveral criteria in conjunction, viz.: (a) location of maxima at the wavelengths typical of E. huxleyi bloom in Rrs spectra; (b) Rrs absolute value ranges at six wavelengths obtained through a dedicated/ large-size statistical sets of spaceborne data from the six seas; (c) consistency with the results of independent application of the BOREALI hydro-optical algorithm (Korosov et al., 2009; Kondrik et al., 2017a), which through retrieving inter alia the concentration of both coccoliths and chlorophyll-apermits to obtain the spatial distribution of the E. huxleyi bloom. This triple checking assured a higher reliability of the algorithm.

3b. Delta pCO2 retrieval algorithm

[Figure]

Again we believe that it would be improper to give here the respective entire excerpt from the paper on pCO2published in a refereed journal (Kondrik et al., 2018a). In a nutshell:

(i) the algorithm has the accuracy of delta pCO2 retrieval that is characterized by the following statistical parameters r2 = 0.54, pâĽ0.001, and RMSE = 23.4$\mu$atm;

(ii) the ensemble of blue data points in fig. 1 (Kondrik e al., 2018a) that looks like an "enormous spread" were obtained using climatological data and added solely to increase the statistical significance of the regression dependence established through using only in situ data that we could find for our study regions (red dots, their number is 187). Most of these points are within the declared error of 23.4 uatm; the indicated red points do not have the problem of Delta_pCO2 values overestimation indicated by the reviewer. It is also necessary to emphasize that a) "confidence interval" the reviewer refer tois in fact the "prediction limit" while the "confidence limit" has a much smaller variation (about 10 uatm). Also, it is important to be aware that the variation is given in uatm(units of partial pressure), but not in ppmas the reviewer writes.

(iii) all corrections for water temperature were duly conducted using the concurrently collected radiometric and IR satellite data.

(iv) the developed delta pCO2 regression dependence has a truly physical basis. Indeed, the increment of pCO2 in surface water within the E. huxleyi bloom is intimately related to the intracellular production of CO2 through the reaction of calcification and associated generation of coccoliths. The latter are very efficient reflectors of sun light coming into water (just because they don't absorb light but only reflect it). Therefore, the greater the amount of CO2 released through calcification, the more intense the optical signal coming out from the bloom area, especially at the wavelength of Rrs maximum – the parameter in our algorithm that is related to delta pCO2. Incidentally, returning to point 2C in our argumentations above, this is an important difference between coccoliths and diatomic frustules as the latter are not solely reflectors but also

absorbers.

4. The graphical illustration of validation of the retrievals of coccolith concentrations is available in our easily accessible papers published elsewhere, we doubt that the inclusion of those illustrations would be justified.

5. We acknowledge the reviewer's critical remarks in C3 –C4. All necessary changes are entered, the respective references [e.g. Shutler et al. (2010, 2013; Winter et al., 2014] are added to the reference list.

We certainly appreciate the list of references provided by the reviewer although, actually, we were aware of nearly all listed publications. The reason why they were not used is explained in point 1of our answers. As to the worksby Shutler et al. (2010,2013), it is indeed our flaw. We are earnestly grateful to the reviewer for this valuable critical remark.

[Figure]

**Topical Editor Decision: Publish subject to minor revisions (review by editor)** (07 Jan 2019)
by David Carlson
Comments to the Author:
The data description seems thorough and well-organised and I suspect this product will serve many users. Please, however, can the authors attend to a few points:

1) I feel somewhat surprised to see that these authors have ignored two coccolithophore data sets recently published in ESSD - Loveday & Smyth, https://doi.org/10.5194/essd-10-2043-2018 which seems quite relevant to the remote sensing aspects albeit using AVHRR rather than ocean colour, and Daniels et al. https://doi.org/10.5194/essd-10-1859-2018 which addresses the issue of calcification rates. In one of their responses the authors mentioned that the impacts of these blooms on pCO2 represented the unique contribution of this data. But these data also both draw on and contribute to the other two data sets?
Such a comparison may also prove useful for validation (e.g. see https://doi.org/10.5194/essd-10-2275-2018).

2) Please also attend to these comments from a third reviewer:

"I found the paper interesting and useful. The authors have made every attempt to validate their products and build on previous work, and to provide a theoretical basis for the algorithms where possible. They have brought together a large body of different types of data to generate their products, and to validate them. They have taken the trouble to re-process some of the OC-CCI data, when it appeared that the masking applied may not have been appropriate for their particular application.

While this paper was under review, another paper on the same topic, but using AVHRR paper, has been published in ESSD (Loveday and Smyth, ESSD 2018). It would be good to refer to this publication. That other attempts have been made recently to address a similar problem in no way deters from the value of the paper under review: the users should be given the opportunity to accept the product that best suits their requirement, and to evaluate the products themselves.

The writing style leaves some problems with the grammar, and some instances where the statements lean towards the hyperbole (especially in the introduction). I do not know if ESSD editors and copy editors can help the authors deal with them?

Assuming that such minor problems with the language can be fixed, I recommend the paper for publication."

Pending appropriate responses, I may also ask one of the reviewers who volunteered to read a revised version to take a quick final look.

Thank you for considering ESSD.

Dear Editor,

In accordance with the suggestions made by all three reviewers, we have revised the text and send you the resultant version of our paper.

We also noted brief descriptions and links to global multiyear databases from Loveday & Smith 2018 and PIC from NASA Ocean Color in the article.

With our cordial regards

Eduard Kazakov in the name of the paper's authors.

[revised manuscript text omitted]